# Maslinic Acid: A New Compound for the Treatment of Multiple Organ Diseases

**DOI:** 10.3390/molecules27248732

**Published:** 2022-12-09

**Authors:** Yan He, Yi Wang, Kun Yang, Jia Jiao, Hong Zhan, Youjun Yang, De Lv, Weihong Li, Weijun Ding

**Affiliations:** 1Department of Fundamental Medicine, Chengdu University of Traditional Chinese Medicine, 1166 Liutai Avenue, Chengdu 611137, China; 2Department of Clinical Medicine, Chengdu University of Traditional Chinese Medicine, 1166 Liutai Avenue, Chengdu 611137, China; 3Affiliated Hospital of Chengdu University of Traditional Chinese Medicine, Chengdu 611137, China

**Keywords:** Maslinic acid, extraction, purification, identification, biological activity, pharmacokinetics, treatment of multiple organ diseases

## Abstract

Maslinic acid (MA) is a pentacyclic triterpene acid, which exists in many plants, including olive, and is highly safe for human beings. In recent years, it has been reported that MA has anti-inflammatory, antioxidant, anti-tumor, hypoglycemic, neuroprotective and other biological activities. More and more experimental data has shown that MA has a good therapeutic effect on multiple organ diseases, indicating that it has great clinical application potential. In this paper, the extraction, purification, identification and analysis, biological activity, pharmacokinetics in vivo and molecular mechanism of MA in treating various organ diseases are reviewed. It is hoped to provide a new idea for MA to treat various organ diseases.

## 1. Introduction

Maslinic acid (MA) is widely found in a variety of plants including olive [1,2], loquat leaves [3,4], red dates [5,6], eucalyptus [7], crape myrtle [8], sage [9], plantain [10], prunella vulgaris [11], spiny leaf dong [12], etc. Olive oil is rich in MA. Olive oil is the traditional edible oil of people in Mediterranean coastal countries, with good nutritional and health care effects. In addition, it is also widely used in beauty care, which also shows that MA is very safe for the human body. The research shows that the MA content of extra virgin olive oil in different regions is 20–98 mg/kg, the MA content of high acid value olive oil in different regions is 212–356 mg/kg, the MA content in olive residue is the highest, and the MA content in olive residue in different regions is 212–1485 mg/kg [13]. Therefore, olive pomace is often used as raw material for MA extraction.

Pure MA is a white powder, molecular formula: C_30_H_48_O_4_, molecular weight: 472.70, melting point: 266–268 °C. It is insoluble in water and petroleum ether, soluble in ethanol and methanol, and soluble in ethyl acetate, benzene and chloroform. The molecular formula of MA is shown in Figure 1. Research has shown that high concentration of MA does not cause any discomfort or death in the treatment of animal disease models [14]. Therefore, MA is considered as a natural compound with low toxicity and side effects.

In the past few decades, people have become more and more interested in natural compounds with health benefits. MA has been extracted from natural plants, such as jujube, olive and hawthorn [15]. Subsequently, researchers studied its pharmacological activity and potential molecular mechanism for disease treatment and toxicological effect [15], and found that MA has beneficial effects in medicine [15,16], health food [15,16,17], animal feed [18,19] and other fields. The biological activity of MA has been systematically summarized by scholars, but the extraction of MA and the therapeutic mechanism of MA for various organ diseases have not been discussed [15]. In this paper, we will comprehensively and systematically review the research progress on the extraction, isolation, identification, bioavailability and bioactivity of MA, and discuss the preventive and therapeutic effects of MA on diseases related to brain, lung, heart, stomach, intestine and kidney organs. The purpose of writing this review is to provide useful experimental data and research ideas for researchers interested in MA research.

## 2. Extraction of MA

### 2.1. Extraction

Solvent extraction technology is the most classical and convenient extraction method, and also the most commonly used method for extracting natural ingredients. The results have shown that MA [20,21,22,23,24] can be successfully extracted from olive, apple, jujube and other plants with methanol, ethanol and ethyl acetate. Fernandez-Pastor et al. [24] determined that the optimal conditions for extracting MA from olive were as follows: “at T = 65–70 °C, t = 30 min, ethyl acetate (EtOAc) was used as the solvent, and the ratio of material to liquid was 1:40, 1:20 and 1:10 respectively”. The solvent extraction of MA has the disadvantages of time-consuming, high-energy consumption, low extraction efficiency and destruction of biological activity. Therefore, it is necessary to develop an economical and effective MA extraction method.

In order to improve the extraction efficiency of MA, microwave assisted extraction (MAE) is widely used in MA extraction [20,24]. Compared with traditional solvent extraction, MAE has the advantages of low organic solvent consumption, high recovery and low cost [24]. Xie et al. [20] determined the best extraction conditions of microwave assisted extraction: “ethanol as the extraction solvent, extraction time of 5 min, liquid–solid ratio of 30 mL/g, extraction temperature of 50 °C, and microwave power of 600 w”. Compared with traditional solvent extraction, MAE not only greatly shortened the extraction time of MA, but the extraction solution was heated evenly and rapidly, so the extraction efficiency of MA was greatly improved.

In addition, in order to improve the yield of MA and optimize the experimental conditions for MA extraction, ultrasonic-assisted extraction was developed and used. Wu et al. [25] successfully extracted MA from a variety of Chinese medicinal materials by using ultrasonic-assisted dispersion liquid–liquid microextraction (UA-DLLME) with chloroform as the extraction solvent and acetone as the dispersion solvent. Gómez-Cruz et al. [26] used ultrasound-assisted extraction for 16 min to extract MA from olive pomace. Song et al. [27] determined the optimal conditions for ultrasonic assisted extraction of MA from jujube to be: “temperature 55.14 °C, ethanol concentration 86.57%, time 34.41 min, liquid–solid ratio 39.33 mL/g, and the average content of MA extracted was 265.568 (µg/g DW)”. Xie et al. [20] determined the optimal conditions for ultrasonic assisted extraction of MA from olive residue to be: “extraction temperature 50 °C, liquid–solid ratio 30 (mL/g), ultrasonic frequency 60 kHz, extraction 5 min, and ultrasonic intensity 135.6 W/cm^2^”. Goulasal et al. [28] determined that ethanol or methanol–ethanol mixture is the most effective solvent for ultrasonic extraction of MA from olive dregs. Ultrasound-assisted extraction is not only convenient, simple and time-consuming, but also allows extraction at a lower temperature, which can avoid the risk of damaging MA activity at high temperatures. In recent years, supercritical fluid extraction (SFE) with both liquid and gas properties has also emerged. It has the characteristics of high diffusivity, low viscosity and good solubility. The experimental results show the optimum conditions for supercritical extraction of MA to be: “CO_2_ as the medium, extraction pressure 22 MPa, extraction temperature 60 °C, material liquid ratio 1:40, extraction time 3 h”. Under these conditions, the extraction yield of MA was the highest [29,30,31].

Solvent extraction, microwave-assisted extraction, ultrasonic-assisted extraction and supercritical fluid extraction are compared. Solvent extraction of MA has the disadvantages of time-consuming, high-energy consumption, low extraction efficiency and destruction of biological activity. MAE greatly shortens the extraction time of MA, and the extraction solution is heated evenly and rapidly. Ultrasound-assisted extraction is not only convenient, simple and time-saving, but also can be performed at a lower temperature, which can avoid the risk of destroying MA activity at high temperatures. SFE is the only method that does not consume organic solvents, but is time-consuming and inefficient.

### 2.2. Purification and Separation of MA

The separation and purification procedures are summarized as follows: In short, the MA liquid extract is first concentrated, and then dried until the weight of the solid sediment remains unchanged, so that the MA crude extract can be obtained. The crude extract of MA was extracted with ethanol (100% *v*/*v* ethanol, 24 h, 10% *w*/*v* solid load) to obtain ethanol extract [32]. Finally, the ethanol extract was purified by HPLC. Water containing 5% acetonitrile (A) and 100% acetonitrile (B) was used as the mobile phase on the chromatographic column for gradient elution. The flow rate of the mobile phase was 1 mL min^−1^, and the column temperature was set at 30 °C [25]. It should be noted that in order to prevent bubbles from escaping from the system and affecting the mobile phase, the mobile phase used for HPLC needs to be ultrasonically treated in advance to be completely degassed [33]. The elution procedure is as follows: 0–5 min, 65–90% B; 5–15 min, 90–92% B; 15–35 min, 92–92.5% B; 35–38 min, 92.5–100% B. The final steps include recrystallization with methanol, filtration, freeze drying and finally obtaining purified MA.

### 2.3. Identification of MA

#### 2.3.1. Colorimetric Method

MA is a pentacyclic triterpene acid compound with 2,3 dihydroxy groups, which has the characteristic color reaction of terpenoids. MA is a 12-alkene structure with slow color development. The main color reaction is as follows: (1) MA reacts with acetic anhydride concentrated sulfuric acid: the sample is dissolved in acetic anhydride, then a few drops of concentrated sulfuric anhydride (1:20) are added, which can produce yellow → red → purple → blue and finally fade, and this color change could identify the sample as MA. (2) MA reacts with trichloroacetic acid: the sample chloroform solution is dropped on filter paper, then 1 drop of 25% TCA solution is added and heated to 100 °C for color development; if red to purple color was generated, this color change could identify the sample as MA. (3) MA reacts with chloroform concentrated sulfuric acid: the sample is dissolved in chloroform. After adding concentrated sulfuric acid, chloroform is red or blue, and the sulfuric acid layer is green fluorescent, so the sample can be identified as MA. (4) MA reacts with antimony pentachloride: the chloroform or ethanol solution of the sample is dropped on the filter paper, and then 20% antimony pentachloride chloroform solution is sprayed on the filter paper. If it is heated, it will show blue, gray blue and gray purple color spots, and this color change could identify the sample as MA. (5) Glacial acetic acid–acetyl chloride reaction: The sample is dissolved in glacial acetic acid, and then a few drops of acetyl chloride and a few grains of zinc oxide are added. After the sample solution is slightly heated, if the color changed to light red or purplish red, this color change could identify the sample as MA [34,35].

#### 2.3.2. Chromatography

In addition to colorimetric methods, we can also use chromatographic methods to identify MA. Guo et al. [6] used high-performance liquid chromatography (HPLC) coupled with evaporative light-scattering detection (ELSD) to identify MA in date palm leaves. Yang et al. [36] also analyzed the content of hawthorneic acid in plum blossoms using high-performance liquid chromatography coupled with evaporative light-scattering detection (HPLC-ELSD). Perez-Camino et al. [13] used gas chromatography (GC) to alkalize MA samples, which were then injected into GC to obtain gas chromatograms of MA alkylated derivatives, and this method was used to identify MA in olive oil. In GC, MA cannot be injected directly into the chromatograph due to its low volatility and high molecular weight, and must be derivatized prior to gas chromatographic analysis. The use of HPLC hinders the detection of MA because it has a saturated backbone, does not show fluorescence and has a very low UV absorption. Therefore, this method may not be sensitive enough to detect low doses of hawthorneic acid. Sánchez-González et al. [37] developed a new detection technique, liquid chromatography–atmospheric pressure chemical ionization mass spectrometry (LC-APCI-MS), for the analysis of MA in plasma. Compared to GC, LC-APCI-MS does not depend on specific groups in the molecule, thus reducing the derivatization step and avoiding sample consumption. Also, it can be used for highly specific detection without long chromatographic separations compared to HPLC. Therefore, LC-APCI-MS may be a relatively effective and sensitive method to determine the concentration of MA. In conclusion, the extraction, purification and identification of MA are shown in Figure 2.

## 3. Biological Activity of MA

### 3.1. Hypoglycemic Effect

MA has been widely studied in reducing blood glucose. As a hypoglycemic drug, MA has the following advantages: (1) non-toxic, fewer side effects; (2) treatment of various complications of diabetes, such as nonalcoholic fatty liver disease caused by diabetes [38], diabetes nephropathy [39] and other complications, which seriously endanger human health. In addition, MA can also act as an inhibitor of glycogen phosphorylase, inhibit the activity of glycogen phosphorylase, reduce glycogen decomposition and reduce blood sugar [40]. It has also been shown that MA can reduce the activity of carbohydrate hydrolase and glucose related transporters in the small intestine during the digestion and absorption of carbohydrates in the small intestine, and reduce the absorption of glucose [41]. After a large amount of glucose intake, MA can also increase insulin sensitivity by activating the AMPK/Sirtl pathway, and promote the transformation and storage of blood glucose [38,40,42].

### 3.2. Antioxidant Effect

There are two kinds of antioxidant systems in the body. One is the non-enzymatic antioxidant system, including ergot thiophene, vitamin C, vitamin E, glutathione and trace elements. The other is the enzyme antioxidant system, including CAT, SOD and GPX. Oxidative stress is a negative effect of free radicals in the body and is considered to be one of the important factors leading to aging and disease [43]. Free radicals and lipid peroxides produced in the process of oxidative stress can cause cross-linking and polymerization of life macromolecules such as proteins and nucleic acids, leading to protein damage, inhibition of RNA synthesis and damage to normal cell structure, which may interfere with cell cycle, chromosome separation, hormone synthesis, cell proliferation and apoptosis, and even lead to dysfunction of oocytes [44]. It is reported that MA has an antioxidant effect and can improve oxidative damage caused by oxidative stress [31]. It was found that MA mainly prevented the generation of oxidative stress by preventing oxidative stress caused by excessive H_2_O_2_, thereby reducing the production level of ROS [31], inhibiting the ability of low-density lipoprotein (LDL) peroxidation [45] and inhibiting the production of nitric oxide (NO) induced by lipopolysaccharide (LPS) [46]. However, it is worth noting that although MA has antioxidant properties, MA can lead to an increase in reactive oxygen species in cells at higher doses, further inducing cell damage and apoptosis [47]. The results showed that in MA-treated B16F10 melanoma cells, ROS levels were decreased at MA concentrations less than (IC_50/8_) and increased at MA concentrations greater than (IC_50/8_) [48].

### 3.3. Neuroprotective Effect

The nervous system plays a leading role in maintaining the homeostasis of the internal environment, maintaining the integrity and unity of the body, and its coordination and balance with the external environment. When the nervous system is injured, a series of diseases will appear, such as cerebrovascular disease, peripheral neuropathy, spinal cord disease, extrapyramidal disease, neuromuscular junction disease, paroxysmal disease, etc., which not only damage human health, but also bring a heavy psychological and economic burden to patients’ families and society. Some studies have shown that MA can significantly reduce ischemia-induced neuronal apoptosis, reduce the level of MDA, and increase SOD activity in the cortex and hippocampus; MA can inhibit NF in the astrocyte κB signal pathway [49]. MA can also reduce the glutamate toxicity of primary neurons in the cerebral cortex in a dose-dependent manner, and improve the survival rate of neurons [50]. At the same time, MA can also regulate extracellular glutamate concentration by increasing the expression of glutamate transporter in astrocytes, which may provide neuroprotection [50]. After the occurrence of nerve injury, MA can also promote Akt activity, increase the growth of nerve axons and synapses, and restore the loss of nerve synapses [48].

### 3.4. Anti-Inflammatory Effect

Inflammation refers to the pathological process in which the body mainly defends against the stimulation of inflammatory factors. Acute inflammatory reaction is one of the most active and important molecular processes in the human body. It clears damaged tissues, restores homeostasis, and protects the host from the threat of pathogens or damage. It is now believed that uncontrolled inflammation is pathological and related to many diseases, such as cardiovascular disease, asthma, neurodegenerative disease, diabetes and obesity [51]. Although the current anti-inflammatory treatment can treat the uncomfortable signs and symptoms caused by inflammation, it may lead to severe immunosuppression and opportunistic infection. Natural compounds have become the focus of anti-inflammation because of their extensive biological activity, low toxicity and weak side effects. Interleukin-1β (IL-1β) is one of the key mediators in stimulating inflammatory response. It was found that in IL-1β-induced inflammatory reaction, MA can effectively pass the PI3K/AKT/NF-κ channel B [52], HMGB1/TLR4/NF-κB pathway [53] and p-STAT-1 pathway [54], inhibit the expression of various inflammatory factors and thus inhibit the inflammatory response.

### 3.5. Anti-Tumor Effect

At present, radiotherapy, chemotherapy and surgery are the main treatments for tumors. However, some advanced and well differentiated patients have poor sensitivity to radiotherapy, while conventional chemotherapy drugs have large side effects and are prone to drug resistance, which significantly reduces the quality of life of patients. Traditional Chinese medicine (TCM) has many advantages in treating tumors, such as rich drug sources, flexible prescriptions and small side effects. It has become one of the research hotspots of anti-tumor drugs to search for anti-tumor active components from traditional Chinese medicine and then study their mechanism of action. Studies have found that MA has a wide range of anti-tumor activities. MA has good anti-tumor activity in a variety of tumors, including colon cancer [55,56,57,58], lung cancer [59,60], colorectal cancer [61,62], pancreatic cancer [59,63,64], kidney cancer [65], prostate cancer [66], bladder cancer [67], gallbladder cancer [68], papilloma [69], gastric cancer [70], human neuroblastoma [71], cardiac cancer [72], etc. The mechanisms of MA in treating tumors include inhibiting cell proliferation [63], blocking cell cycle [48,73], inducing cancer cell apoptosis [56] and inhibiting angiogenesis [74]. The biological activity of MA is summarized in Figure 3.

## 4. Bioavailability of MA

Although MA has so many biological activities, its in vivo efficacy in the human body also depends on its absorption and metabolism, so it is important to understand the bioavailability of MA. Bioavailability is the fraction of an orally administered dose of intact drug that reaches the systemic circulation, taking into account absorption and local metabolic degradation. The NUTRAOLEUM study [75] was used to evaluate the pharmacokinetics of MA in the human body in a single dose of high triterpene-acid content olive oil. After repeated administration of MA, its plasma concentration gradually accumulated in the dry period of one week. After a single dose of 30 mL MA for 24 h, it increased by 6.7 ng/mL from 1.8 ng/mL at baseline, and reached 21.5 ng/mL three weeks later. The bioavailability based on Cmax and AUC0-10 is 7 times higher than that of OA. The oral bioavailability of MA is about 6.25%.

Studies in rodents have shown that [75] MA is rapidly absorbed orally, reaching a peak at 0.51 h with a bioavailability of 5.13%, which is consistent with the reported values for other pentacyclic triterpenoids; after entering the blood MA was then widely distributed in tissues with central and peripheral distribution volumes of 8.41 L/70 kg and 63.6 L/70 kg, respectively. The reason for the low oral availability of MA may be that MA is affected by the mechanical action of the digestive tract, digestive enzymes, pH and other foods consumed together during oral administration. After oral administration of triterpenes for 4 weeks, the concentration of triterpenes in organs was measured, and the results showed that some triterpenes including MA had the highest concentrations in the liver, followed by the kidneys [76]. It is suggested that the liver may be the main organ for MA storage. Furthermore, MA is able to penetrate the blood–brain barrier after being absorbed and metabolized [76], a finding that supports the possibility of using MA for brain protection.

## 5. Therapeutic Effects of MA

### 5.1. MA Treatment of Brain Diseases

The therapeutic effect of MA on brain diseases is mainly through anti-inflammatory [49], antioxidant [77] and neuroprotective bioactivity [50]. MA is mainly used to treat Alzheimer’s disease (AD), epilepsy, stroke, astrocytoma, etc.

#### 5.1.1. MA Treatment of Alzheimer’s Disease

AD is a neurodegenerative disease whose pathogenesis is still unclear. It is currently believed that impaired brain neurotrophic factor (BDNF) signaling may be one of the pathogenic mechanisms leading to neurodegenerative diseases [78]. BDNF is a neuroprotective agent that is very important for neuronal survival and neurotransmission, and can prevent and improve neurodegeneration in AD [78]. A study has shown that BDNF can prevent and improve memory cognitive impairment in AD [79]. In addition, BDNF has been reported to be critical for memory consolidation and enhancement 9 to 12 h after memory acquisition [80,81]. Bae et al. [82] used a scopolamine-induced mouse AD model and then treated the mice with MA (concentration of 1 mg/kg and 3 mg/kg). The results showed that the mRNA expression level of BDNF was increased, neuronal apoptosis was reduced, and the cognitive function and memory cognitive impairment of the mice were improved 9 and 12 h after treatment. Thus, MA may be a potential drug for the treatment of AD.

#### 5.1.2. MA Treatment of Epilepsy

Epilepsy is a chronic brain dysfunction caused by excessive discharge of neurons in the brain, and both experimental and clinical evidence suggests that inflammatory stress in the brain is associated with epilepsy, especially in the hippocampus [83,84]. Key inflammatory mediators contribute to the progression of epileptic seizures, such as IL-1β, IL-6, TNF-α and prostaglandin E2 (PGE2) [85,86]. Excessive production of reactive oxygen species (ROS) can also lead to seizures associated with neuronal depolarization and exacerbate neurological dysfunction [87]. In addition, glutamate excitotoxicity is another important factor that induces epilepsy [88]. Therefore, any drug that reduces inflammation, oxidation or glutamate toxicity in the hippocampus can reduce the severity of epilepsy.

Wang et al. [89] used kainic acid (KA) to induce epileptiform behavior in mice, which were at the same time given 20 or 40 mg/kg/d MA (MA) treatment; the results showed that MA preadministration reduces the production of IL-1β, IL-6, TNF-α and PGE2, which are inflammatory factors. Also, MA preadministration maintains the activity of glutathione peroxide (GPX) and glutathione reductase (GR), thereby reducing oxidized glutathione (GSSG) production and preserving glutathione (GSH), reducing ROS production, enhancing hippocampal glutamine synthetase activity, decreasing glutamate levels, and increasing glutamine levels, thereby alleviating epileptiform behavior. These findings suggest that MA may be an effective therapeutic agent for seizure relief.

#### 5.1.3. MA Treatment of Ischemic Stroke

Ischemic stroke is a cerebrovascular disease that causes ischemia and hypoxia in brain tissue due to impaired blood supply to the brain, resulting in necrosis, softening and eventually infarct foci. Neural regeneration and brain repair in the injured area after ischemic stroke mainly rely on axonal remodeling and dendritic plasticity. MA has a better preventive effect on ischemic stroke. Qian et al. [90] induced an ischemic stroke model by oxygen–glucose deprivation, and the experimental results showed that MA treatment normalizes the expression/activation of caspase3 and caspase9, and increases the ratio of Bcl-2/Bax. In addition, MA inhibited NO production and iNOS expression induced by oxygen–glucose deprivation. These results indicate that MA has beneficial effects on hypoxic neurons by inhibiting the activation of iNOS. It indicates that MA has a better preventive effect on ischemic stroke and can reduce the necrosis of neurons. Guan et al. [91] found that MA also has a good therapeutic effect on the stage of stroke. MA treatment can enhance the expression of glial glutamate transporter GLT-1 at the protein and mRNA levels, leaving extracellular glutamate at a low concentration, thus playing a protective role in nerve cells during the ischemic stroke. In addition, Qian et al. [92] found that the presence of MA prolonged the therapeutic time window of MK-801, and the combination of the two may be more effective in the treatment of acute ischemic stroke. For the recovery period of stroke, MA can also have a therapeutic effect. After MA was given to mice treated with reperfusion after ischemic stroke for a period of time, it was found that synaptophysin levels increased significantly, nerve axon damage recovered significantly, and synapse growth and synaptogenesis increased [93]. The above experimental results indicate that MA has therapeutic effects on ischemic stroke through multi-stage and multi-pathway effects.

#### 5.1.4. MA Treatment of Malignant Astrocytoma

Malignant astrocytoma is a common primary brain tumor. Traditional Chinese medicine has many advantages in the treatment of tumors, such as rich sources of drugs, flexible prescriptions and small side effects. It has become one of the research hotspots of anti-tumor drugs to find anti-tumor active components from traditional Chinese medicine and study their mechanism of action. It has been found that MA has a good therapeutic effect on malignant astrocytoma.

Martín et al. [77] incubated 1321N1 cells with different concentrations of MA (1, 5, 25 and 50 μmol/L) for 24 h. It was found that MA led to an increase in intracellular reactive oxygen species, loss of mitochondrial membrane integrity and subsequent apoptosis of tumor cells. Therefore, MA may be a promising new drug for the treatment of astrocytoma.

### 5.2. MA Treatment of Lung Diseases

MA can treat lung diseases through various anti-inflammatory pathways [54,94] and, at the same time, it can also treat lung cancer by inhibiting cancer cell proliferation and inducing cancer cell apoptosis [59,60].

#### 5.2.1. MA Treatment of Lung Cancer

Lung cancer is one of the most common cancers in the world. After many treatments, the five-year survival rate of lung cancer is still very low [95]. Therefore, it is very important to find drugs that can induce apoptosis of lung cancer cells.

Hsia et al. [59] incubated human lung cancer A549 cells with different concentrations of MA (4, 8, 16, 32 and 64 μmol/L) and found that MA can mediate the apoptosis pathway of tumor cells and HIF-1 α, and cause apoptosis of cancer cells. Bai et al. [60] incubated human lung cancer A549 cells with different concentrations of MA (0, 9, 12, 15, 18, and 21 µg/mL) for 24 h, and found that MA can regulate Smac expression and reduce c-IAP1, c-IAP2, XIAP and survivin expression to regulate the expression of caspase-3, caspase-8 and caspase-9, thereby inducing cancer cell apoptosis. These findings prove that MA can induce apoptosis of lung adenocarcinoma cells, and thus have a therapeutic effect on lung cancer.

#### 5.2.2. MA Treatment of Lung System Damage

Lung injury is lung parenchyma injury caused by different factors, and its pathological mechanism is mainly related to oxidative stress and inflammatory injury. Therefore, anti-inflammatory and antioxidant drugs may have beneficial effects on lung injury. Jeong et al. [94] induced lung injury in mice using PM2.5 (10 mg/kg) generated by intratracheal instillation of diesel and, 30 min later, MA (0.2, 0.4, 0.6 and 0.8 mg/kg) was injected via the tail vein. It was found that MA has the ability to modulate the TLR4-MyD88 and mTOR autophagy pathways to counteract PM2.5-induced lung injury. Lee et al. [54] injected LPS (15 mg/kg i.p.) into the peritoneal cavity to induce lung injury, followed by intravenous injection of MA (0.07–0.7 mg/kg) six hours after the injection, and found that MA can down-regulate NF-κB and p-STAT-1 to regulate iNOS, exerting an anti-inflammatory effect. These results suggest that MA may be a potential natural compound for the treatment of lung injury.

### 5.3. MA Treatment of Heart Disease

MA plays a therapeutic role in heart-related diseases through its anti-hyperlipidemia, inhibition of lipid peroxidation (LPO), and anti-blood-glucose and antioxidant effects.

#### 5.3.1. MA Treatment of Pathological Cardiac Hypertrophy

Physiological cardiac hypertrophy is an adaptive response to physiological and pathological stimuli and is designed to cope with increased workload [96]. However, pathological cardiac hypertrophy is a major risk factor for cardiomyopathy, heart failure and sudden cardiac death [97]. The mechanism of pathological myocardial hypertrophy is that cardiomyocyte hypertrophy enters a decompensated phase, resulting in cell death, myocardial fibrosis, sarcomere structure changes, metabolic reprogramming, protein damage, etc., which ultimately lead to poor cardiac remodeling and heart failure [98]. N6-methyladenosine (m6A) is the most prevalent internal modification of mammalian messenger RNA (mRNA) [99], and m6A plays an important role in cardiac biological processes and the pathogenesis of cardiovascular disease [100,101]. At the same time, there is increasing evidence that the extracellular regulated protein kinase (ERK) and protein kinase B (PKB/AKT) signaling pathways are involved in the process of cardiac hypertrophy [102,103]. Therefore, pharmacological intervention with respect to these molecules could facilitate the development of treatments for cardiac hypertrophy.

Fang et al. [104] found that MA treatment significantly inhibited the hypertrophy of NMCMs induced by Ang II, and the dose did not affect the cell viability of NCMCs. At the same time, the researchers established a mouse model of transverse aortic contraction (TAC) to simulate cardiac hypertrophy caused by pressure overload, and then injected MA (30 mg/kg/d) intraperitoneally for 14 days. MA administration reduced the level of total RNA m6A methylation and METTL3 in TAC stressed hearts. In vivo and in vitro, MA can significantly improve myocardial hypertrophy, myocardial fibrosis and cardiac function. Liu et al. [105] ligated the aorta of mice to induce cardiac hypertrophy caused by pressure overload. After operation, the mice took MA (20 mg/kg) orally for 4 weeks, and the hypertrophic cardiomyocytes induced by PE were treated with MA in vitro. It was found that MA alleviated the pressure-overload-induced cardiac hypertrophy in vivo and PE-induced cardiac hypertrophy in vitro by reducing the phosphorylation of AKT and ERK signaling pathways. These results suggest that MA may be a potential drug for the treatment of cardiac hypertrophy.

#### 5.3.2. MA Treatment of Acute Myocardial Infarction

Ischemic heart disease has become one of the most important causes of death worldwide [96]. Abnormal lipid metabolism is the most important risk factor for atherosclerosis, and high triglycerides are an independent marker of increased risk of ischemic events [106]. Oxyphosphatase (PON) prevents oxidation of low-density lipoprotein cholesterol (LDL-C) and protects cell membranes from free radical damage to prevent atherosclerosis [107], and is an important target for the treatment of ischemic cardiomyopathy. In addition, xanthine oxidase (XO) is an enzyme that produces free radicals, which can produce superoxide, hydrogen peroxide, NADH and NO, and eventually lead to some cardiovascular diseases (CVD). Therefore, XO is a potential drug target for the treatment of CVD [108].

Hussain Shaik et al. [109] used MA (15 mg/kg) to intervene in a myocardial infarction model induced by isoproterenol (85 mg/kg). It is found that MA can increase PON activity, reduce LDL-C level and inhibit LPO activity, thus playing a protective role in the heart. Another experiment by Shaik et al. [110] found that MA can relieve myocardial infarction (MI) by inhibiting XO. Therefore, MA can be considered as a potential natural drug for the treatment of MI without obvious adverse reactions.

#### 5.3.3. MA Treatment of Diabetic Heart Disease

Diabetes is a global epidemic with a strong association with cardiovascular disease [111]. Diabetes heart disease is due to the imbalance of sugar and lipid metabolism related to diabetes, which leads to increased oxidative stress and activation of multiple inflammatory pathways, ultimately leading to tissue damage, heart remodeling, heart dysfunction, etc. [112].

Khathi et al. [41] found that the IC_50_ values for sucrase, α-amylase and α-glucosidase were lower for different concentrations of MA than for acarbose, suggesting that MA was effective in controlling blood glucose. Hung et al. [113] injected streptozotocin (50 mg/kg) into the tail vein of mice to induce diabetes in mice (fasting blood glucose level ≥ 200 mg/dL), and then fed mice with 0.1% MA and 0.2% MA diet for 12 consecutive weeks to study the cardioprotective effect of MA in diabetic mice. This study found that MA can protect the hearts of diabetic mice by alleviating glycosyl damage and coagulation disorders, inhibiting NF-κB and MAPK pathways, and reducing oxidative stress. While inhibiting oxidative stress, MA can also regulate blood sugar and improve heart function without obvious adverse reactions. MA can be used to treat diabetes heart disease [19].

### 5.4. MA Treatment of Liver Diseases

MA effectively inhibits fat synthesis and restores hepatic glycogen to normal levels [114,115]. In addition, MA increases the activity of antioxidant enzymes through anti-inflammatory and antioxidant effects during liver injury, improves liver function [38], and can also reduce the production of inflammatory factors and protect the liver. Finally, MA can also inhibit the metastasis and invasion of liver cancer [116].

#### 5.4.1. MA Treatment of Acute Liver Injury

Acute liver injury is a life-threatening syndrome with high morbidity [117], which is often caused by viruses, bacteria, drugs and toxins [118,119], and the pathological mechanisms involved are complex and yet to be elucidated. However, there is increasing evidence that inflammation and oxidative stress are involved in the pathogenesis of liver injury [120]. For example, the detrimental effects of alcohol in alcoholic acute liver injury are mainly attributed to the large production of reactive oxygen species (ROS) and acetaldehyde in ethanol metabolism, which deplete glutathione (GSH) and lead to free-radical-mediated apoptosis [121,122]. In addition, ethanol and its metabolites enhance the formation of inflammatory cytokines such as IL-6 and TNF-α [123,124], partly due to increased stimulation of oxidative stress, which can lead to cellular factor imbalances and immune disorders, and further impair liver function. Therefore, any drug that can inhibit inflammation and oxidative stress has the potential to reduce liver damage [125].

Yan et al. [126] found that MA can significantly inhibit CYP2E1 and NF-κB and MAPK pathways, thereby reducing downstream oxidation and inflammatory factors (such as NO, TNF- α And PGE2, etc.), and finally reduce alcohol induced liver injury. This indicates that MA can effectively prevent alcohol-induced acute liver injury. Wang et al. [127] gave MA before LPS/D-gal-induced liver injury, and found that MA inhibited NF-κB and activate the Nrf2 signaling pathway to play an anti-inflammatory and antioxidant role, thereby reducing liver damage. Relevant experiments show that MA can reduce liver injury induced by various factors through anti-inflammatory and antioxidant effects, so these findings may support MA as an effective preventive agent for acute liver injury.

#### 5.4.2. MA Treatment of Liver Cancer

Liver cancer, also known as hepatocellular carcinoma (HCC), is a hypervascular tumor characterized by massive angiogenesis [128]. It is reported that hypoxia is the main pathophysiological condition to promote angiogenesis, and hypoxia-inducible factor (HIF-1R) regulates the basic adaptive response of cancer cells to hypoxia. Vascular endothelial growth factor (VEGF) [129], urokinase plasminogen activator (uPA) [130], reactive oxygen species (ROS) and nitric oxide (NO) [131,132] play an important role in cancer angiogenesis. Therefore, any drug that can inhibit angiogenesis may be beneficial to the treatment of HCC.

Lin et al. [116] cultured hepatoma Hep3B, Huh7 and HA22T cells in vitro, and treated these hepatocellular carcinoma cells with 2 or 4 μmol/L MA for 72 h. It was found that 4 μmol/L MA significantly inhibited the mRNA expression of angiogenic factors HIF-1R, VEGF, IL-8 and uPA, and reduced ROS by maintaining GSH levels and reducing NO production, while MA also reduced the invasion and migration of three types of hepatocellular carcinoma cells. Thus, MA is an effective anti-angiogenic agent that delays the invasion and migration of hepatocellular carcinoma cells.

#### 5.4.3. MA Treatment of Nonalcoholic Fatty Liver Disease

Nonalcoholic fatty liver disease develops from abnormal lipid metabolism. Excessive intake of free fatty acids interferes with lipid storage and metabolism, and also forms excessive accumulation of triglycerides and lipids in the liver, eventually leading to chronic hepatitis with abnormal liver function and liver metabolic disorders [133]. If patients fail to maintain an appropriate body weight and a balanced diet, the disease may progressively worsen and progress to nonalcoholic steatohepatitis, irreversible fibrosis or cirrhosis, and even hepatocellular carcinoma [134].

Liou et al. [114] induced nonalcoholic fatty liver disease in mice by feeding mice with HFD (60% fat, *w*/*w*) for four weeks. After successful modeling, mice were injected intraperitoneally with MA (10 mg/kg and 20 mg/kg) twice a week for 12 weeks. Experiments found that MA can reduce liver fat infiltration by inhibiting the expression of genes involved in liver adipogenesis, and restore liver glycogen levels and reduce triglycerides and total cholesterol by enhancing ATGL and Sirt1 expression, and AMPK phosphorylation. It is suggested that MA can prevent obesity-induced nonalcoholic fatty liver disease by regulating the Sirt1/AMPK signaling pathway.

### 5.5. MA Treatment of Gastric Diseases

MA mainly protects the stomach by enhancing gastric mucosal protective factor [135] and inhibiting the activity of enzymes such as H[+], K[+]-ATP [136].

#### 5.5.1. MA Treatment of Gastric Ulcer

Gastric ulcer is one of the most common gastric diseases in the world, which is mainly related to the destruction of gastric mucosa by aggressive factors (hydrochloric acid, pepsin, bile reflux and reactive oxygen species) [137,138]. Da Rosa et al. [136] pre-administered MA (1–10 mg/kg) to mice and, 1 h after oral administration or 30 min after intraperitoneal treatment, mice were given ethanol/HCl (60%/0.3 M, 10 mL) /kg, po) or indomethacin (80 mg/kg, orally) to induce gastric injury. The results showed that MA pretreatment can effectively reduce the area of gastric injury by more than 90%, and the mechanism may be the inhibition of H[+], K[+]-ATPase activities. This experiment shows that MA can protect gastric mucosa and thus play a therapeutic role in gastric ulcer.

#### 5.5.2. MA Treatment of Gastric Cancer

One study found that MA was able to inhibit IL-6 expression, induce JAK and STAT3 phosphorylation, and down-regulate STAT3-mediated protein Bad, Bcl-2 and Bax expression to treat gastric cancer [70].

### 5.6. MA Treatment of Intestinal Diseases

In the intestine, MA can inhibit carbohydrate hydrolysis and reduce glucose absorption, and can also induce apoptosis in colorectal cancer cells.

#### MA Treatment of Colorectal Cancer

Colorectal cancer (CRC) is one of the five major cancers worldwide, and its conventional treatment is mainly surgery, radiotherapy and chemotherapy [139], but there are problems such as recurrence and significant side effects [140]. Therefore, there is an urgent need to develop a potent drug with low side effects to treat CRC.

In 2006, Reyes et al. [141] found that MA exerts anti-proliferative and pro-apoptotic effects in human colon cancer cell lines HT29 and Caco-2. In recent years, it has been found that, in human colon cancer cells HT29, MA caused G0/G1 phase cell cycle arrest and induced tumor apoptosis through the JNK-Bid-mediated mitochondrial apoptotic pathway and p53 activation [53]. In p53-deficient Caco-2 colon cancer cells, MA was also able to rapidly activate caspase-8 and caspase-3, resulting in late activation of caspase-9 while Bax protein expression levels remained unchanged [56,57]. This suggests that MA can exert anti-tumor effects by activating the endogenous mitochondrial apoptotic pathway or exogenous apoptotic pathway. In addition, MA can also exert anti-tumor effects by inducing cytoskeletal changes in HT29 human colon cancer cells [58]. In vivo experiments have also demonstrated that MA can reduce intestinal tumorigenesis in ApcMin/+ mice by inhibiting the formation of intestinal polyps [142]. It has also been found that MA dose-dependently regulates the AMPK-mTOR pathway, thereby inhibiting SW480 and HCT116 colon cancer cell viability to exert anti-tumor effects [61]. At the same time, MA at high concentrations in the intestine can be useful for better prevention of colon cancer [143]. However, the oral bioavailability of MA is low, about 5% [144]; the relative abundance of MA metabolites in intestinal contents is higher than that in plasma or urine. Therefore, oral MA may be a more effective drug delivery method to prevent colon cancer.

### 5.7. MA Treatment of Kidney Disease

MA protects the kidneys through anti-inflammatory, hypoglycemic and antioxidant effects, and can also inhibit the development of renal cell carcinoma by inhibiting the proliferation and generation of blood vessels.

#### 5.7.1. MA Treatment of Diabetic Nephropathy

Diabetic nephropathy occurs in approximately 40% of people with diabetes [145]. Diabetes nephropathy has the strongest correlation with the death of diabetes patients [146], and has increased the cardiovascular incidence rate and mortality of diabetes patients [147].

Studies have shown that MA inhibits the expression of oxidative stress markers and inflammatory factors in the kidney of diabetic mice, and activates the AMPK/SIRT1 signaling pathway to affect renal metabolism, thereby protecting renal function [42]. Pre-diabetes is the intermediate stage between normal blood sugar and diabetes. The risk of developing into diabetes is very high, and the risk of developing into cardiovascular disease, kidney disease and death will also increase [148]; for example, renal insufficiency can occur at an early stage of impaired glucose metabolism [149]. Based on the significant cost of diabetic complications, it is essential to reverse hyperglycemia and complications prior to pre-diabetes. Preliminary studies have found that MA attenuates renal oxidative stress, reduces urinary podocin mRNA expression and attenuates renal insufficiency [150]. However, molecular mechanisms need to be identified to elucidate the mechanisms by which MA improves renal function. Prevention of diabetic nephropathy should require prevention of renal reabsorption of sodium ions in addition to strict glycemic control. Studies have shown that intravenous infusion of MA derivative phenylhydrazine (PH-MA) significantly increased renal Na+ and lithium excretion fractions, significantly increased glomerular filtration rate (GFR), reduced plasma aldosterone levels and improved diabetic renal function [151]. MA can also reduce the expression of glucose transporter protein 1 (GLUT1) and glucose transporter protein 2 (GLUT2) in the kidneys of diabetic rats when increasing the renal excretion of Na+, thereby reducing the blood sugar level [39]. MA not only reduces the function of renal excretion of Na+, but also can lower blood sugar and improve renal function. MA may be a potential drug for the treatment of diabetes nephropathy.

#### 5.7.2. MA Treatment of Acute Kidney Injury

Acute kidney injury (AKI) is the sudden loss of excretory renal function, mainly due to renal ischemia/reperfusion injury (IRI) following predisposing factors such as renal transplantation and renal surgery [152,153], and leads to renal irreversible damage [154]. After IRI, a large amount of reactive oxygen species (ROS) and calcium overload are produced to activate cell apoptosis, necrosis and necrotic apoptosis, activate inflammatory reaction, and finally lead to renal structural damage and long-term tissue damage [155].

One study showed that MA inhibits IRI-induced AKI injury through NF-κB and MAPK signaling pathways [156]. Inflammation is important for the appearance, progression, exacerbation and prognosis of IRI [157], and MA has a therapeutic effect on IRI-related inflammation, indicating that MA is a promising drug for treating AKI.

Renal cell carcinoma (RCC) is a highly metastatic, heterogeneous disease that is resistant to conventional treatment modalities [155,158]. At present, it is still a refractory cancer, because RCC is a solid tumor with the highest degree of vascularization [159]. MA has antiproliferative and antiangiogenic effects. In metastatic renal cell carcinoma cell lines, MA inhibited the proliferation of cancer cells by reducing nuclear antigen expression, anti-proliferation and anti-colony production in proliferating cells, and down-regulating VEGF in vascular endothelial cells and PCNA in RCC to inhibit angiogenesis and proliferation [65]. It is worth noting that, unlike the mechanism of MA against other tumors, its anti-RCC mechanism of action is mediated by inhibiting proliferation rather than inducing apoptosis. The treatment mechanism of MA in treating diseases of various organs are summarized, as shown in Table 1.

## 6. Conclusions and Future Perspectives

This paper systematically reviewed the biological activity and clinical research progress of MA in recent years. The use of natural compounds to treat diseases has become increasingly popular, because these natural compounds have almost no side effects. As a natural compound, MA has a wide range of biological activities, low toxicity and side effects. As a natural medicine, it has been paid more and more attention in clinical treatment of diseases.

MA has a variety of biological activities such as antioxidant, anti-inflammatory, anti-tumor, hypoglycemic and neuroprotective activities, so it has a good preventive and therapeutic effect on various organ-related diseases, such as some cancers, ischemic diseases, diabetes, stroke, and other acute and chronic diseases caused by oxidative stress or inflammation.

MA is a pentacyclic triterpene acid, which is widely found in medicinal plants such as olive, jujube, hawthorn and loquat leaves. As MA is particularly rich in olive peel, most of the current studies use olive residue to extract MA. MA extraction is usually by the SE extraction method, but it is always time-consuming and inefficient. Although UAE, MAE and SFE can better overcome these problems as new extraction methods, they cannot be widely promoted due to their immature processes and costs.

Although the research on MA has been so comprehensive, there are still many questions to be solved: (1) MA can promote the expression of GLT-1 in astrocytes and reduce the level of extracellular glutamate, but the pathway of MA activating GLT-1 is still unclear. (2) At present, the extraction process of MA is either high energy consumption, time consuming, low efficiency, immature process or high cost, so it is urgent to develop new extraction methods. (3) At present, the in vivo research of MA mostly adopts the method of gavage and intraperitoneal injection. However, clinical medication is generally oral and intravenous infusion, so whether MA can achieve the expected effect through intravenous administration requires further research. (4) MA has antioxidant biological activity, but at higher doses, which in turn exhibits pro-oxidative effects leading to an increase in intracellular reactive oxygen species. (5) MA, as a natural compound, can be used for the prevention and treatment of various diseases, but research into its toxicological effects and side effects under different circumstances is still lacking.

Overall, MA is of great interest in modern scientific research due to its beneficial effects in numerous diseases. MA has a wide range of biological activities, weak side effects and other properties, which makes it have great development potential in the fields of modern farming, functional health care and medical treatment. We believe that the research on MA will become more and more comprehensive, and the practical application of MA will become more and more extensive. It will usher in a broad market and broad prospects, and go out of the laboratory to enter the food and medicine fields.

## Figures and Tables

**Figure 1 molecules-27-08732-f001:**
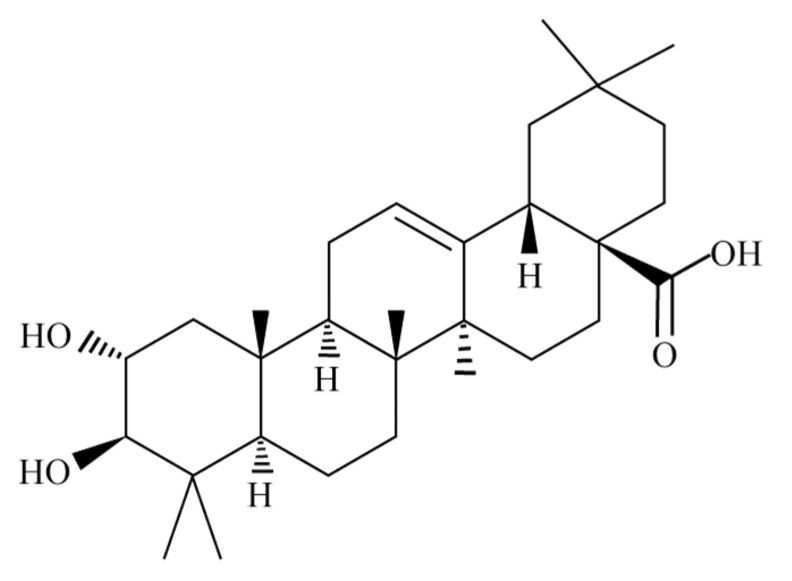
Structural formula of MA.

**Figure 2 molecules-27-08732-f002:**
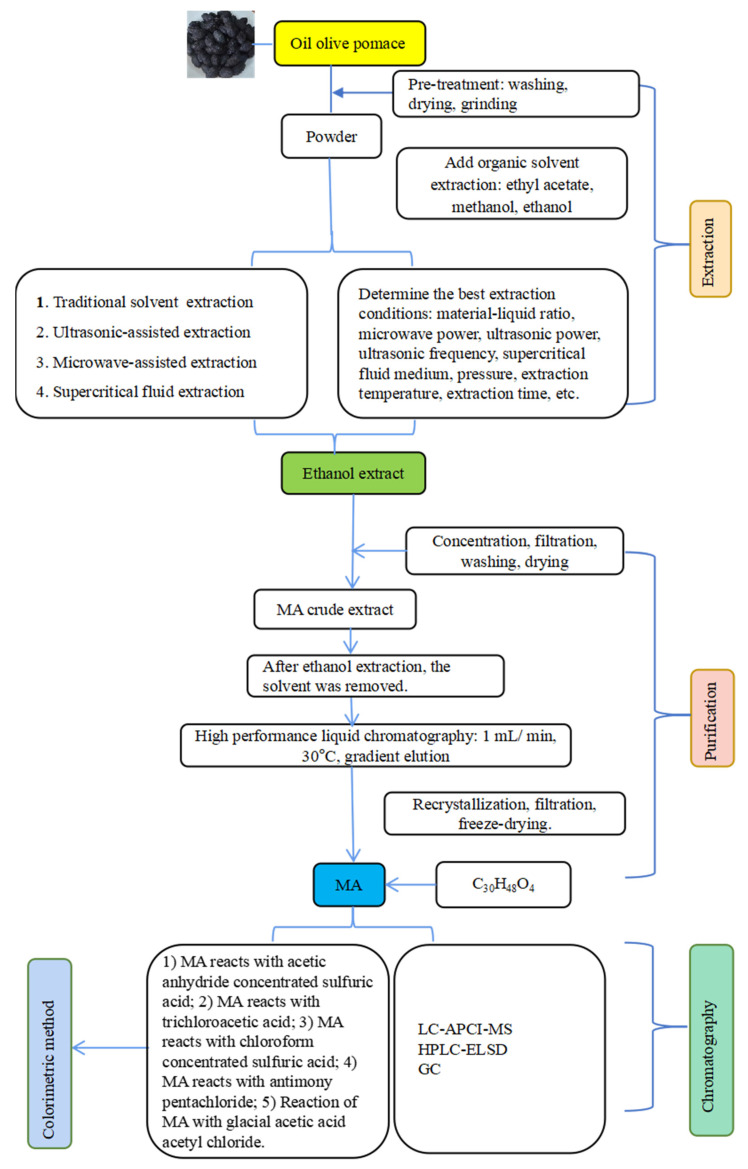
Extraction, purification and identification of MA.

**Figure 3 molecules-27-08732-f003:**
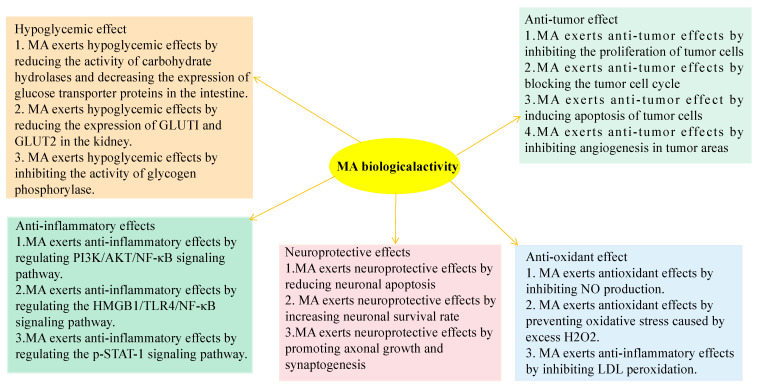
Biological activity of MA.

**Table 1 molecules-27-08732-t001:** The treatment mechanism of MA in treating diseases of various organs are summarized.

Organs	In Vivo/In Vitro	Diseases	Treatment Mechanism	References
Brain				
	In vivo	AD	MA promotes the expression of BDNF, reduces the apoptosis of neurons, improves the memory and cognitive impairment of mice caused by cholinergic system damage, and enhances the cognitive function of mice	[82]
	In vivo	Epilepsy	MA can reduce the production of inflammatory factors, reduce the level of glutamate in the hippocampus, improve the antioxidant capacity of the hippocampus and thus improve the production of epileptic behavior	[89]
	In vitro	Ischemic stroke	MA can block the cell necrosis induced by hypoxia, reduce the necrosis of neurons, effectively prevent the damage of cell bodies and neurites, and increase the survival rate of neurons	[90]
	In vivo	Ischemic stroke	MA prolonged the therapeutic time window of MK-801 from 1 h to 3 h. MA and MK-801 jointly increased the level of glutamate transporter GLT-1 in astrocytes and promoted astrocytes to regulate glutamate excitotoxicity, thus playing a therapeutic role in ischemia	[92]
	In vivo	Ischemic stroke	MA can significantly prevent axon injury, promote axon regeneration and increase the expression of synaptophysin after 7 days of ischemia	[93]
	In vivo	Ischemic stroke	MA treatment can enhance the expression of glial glutamate transporter GLT-1 at the protein and mRNA levels, leaving extracellular glutamate at a low concentration, thus playing a protective role in nerve cells during stroke ischemia	[91]
	In vitro	Astrocytoma (1321N1 cells)	MA can induce apoptosis of 1321N1 cell line	[77]
Lung				
	In vitro	Lung cancer (A549 cells)	MA treatment mediates mitochondrial apoptosis pathway and HIF-1 α pathway induced apoptosis of A549 cells	[59]
	In vitro	Lung cancer (A549 cells)	MA can promote the expression of caspase-3, caspase-8 and caspase-9 by regulating the expression of Smac and reducing the expression of c-IAP1, c-IAP2, XIAP and survivin, thereby inducing apoptosis of A549 cells	[60]
	In vivo	Lung damage	MA antagonizes lung injury caused by diesel PM2.5 by regulating TLR4-MyD88 and mTOR autophagy pathway	[94]
	In vivo	Lung injury	MA exerts anti-inflammatory effects by down-regulating NF-κB and p-STAT-1 to regulate iNOS	[54]
Heart				
	In vitro	Myocardial hypertrophy (NMCMs, H9C2 cells)	MA treatment significantly inhibited Ang-II-induced hypertrophy of NMCMs, and the dose did not affect the cell viability of H9C2 and NCMCs	[104]
	In vivo	Myocardial hypertrophy	MA can significantly improve myocardial hypertrophy, myocardial fibrosis and cardiac function, probably through the METTL3-mediated m 6A methylation pathway	[104]
	In vivo	Myocardial hypertrophy	MA reduces stress-overload-induced cardiac hypertrophy in vivo by reducing phosphorylation of AKT and ERK signaling pathways	[105]
	In vivo	Myocardial infarction	MA provides cardioprotection by increasing PON activity, reducing LDL-C levels and inhibiting lipid peroxidation (LPO)	[109]
	In vivo	Myocardial infarction	MA can inhibit the enzyme xanthine oxidase XO to relieve myocardial infarction	[110]
Liver				
	In vivo	Acute liver injury	MA inhibits CYP2E1, NF-κB and MAPK pathways, reducing the production of downstream oxidative and inflammatory factors (such as NO, TNF-α and PGE2), ultimately reducing alcohol-induced hepatotoxicity	[126]
	In vivo	Acute liver injury	MA exerts anti-inflammatory and antioxidant effects by inhibiting NF-κB and activating the Nrf2 signaling pathway, thereby providing protection against LPS/D-gal-induced liver injury	[127]
	In vitro	Liver cancer (hepatocellular carcinoma Hep3B, Huh7 and HA22T cells)	MA significantly inhibits angiogenesis and delays the metastasis and invasion of liver cancer cells	[116]
	In vitro	Fatty liver disease	MA can reduce hepatic fat infiltration, restore liver glycogen levels and reduce triglyceride and total cholesterol levels by inhibiting the expression of genes involved in hepatic fat formation	[114]
Stomach				
	In vivo	Gastric ulcer	MA pretreatment effectively reduces the area of gastric damage, inhibits H[+] and K[+]-ATPase activity, and provides gastroprotection	[136]
	In vivo	Gastric cancer	MA was able to inhibit IL-6 expression, induce JAK and STAT3 phosphorylation, and down-regulate STAT3-mediated protein Bad, Bcl-2 and Bax expression to treat gastric cancer	[70]
Intestine				
	In vitro	Colorectal cancer (HCT116, SW480 cells)	MA mainly induces apoptosis of colorectal cancer cells and inhibits proliferation and migration of colorectal tumors, and induces apoptosis to play an anti-tumor role	[61]
Kidney				
	In vivo	Diabetic nephropathy	MA activation of renal AMPK/SIRT1 signaling pathway improves diabetic nephropathy	[42]
	In vivo	Diabetic nephropathy	MA increases renal excretion of Na+ and can also lower blood glucose values	[151]
	In vivo	Renal cell carcinoma	MA inhibited the proliferation of cancer cells by reducing nuclear antigen expression, anti-proliferation and anti-colony production in proliferating cells, and down-regulating VEGF in vascular endothelial cells and PCNA in RCC to inhibit angiogenesis and proliferation	[65]
	In vivo	Acute kidney injury	MA inhibits IRI-induced AKI injury via NF-κB and MAPK signaling pathways	[156]

## Data Availability

Not applicable.

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
