# Peer review of "Maslinic Acid: A New Compound for the Treatment of Multiple Organ Diseases"

_molecules, 2022, doi:10.3390/molecules27248732_

Round 1
Reviewer 1 Report
The manuscript submitted by He et al reports the review on Maslinic acid as a potential candidate for the treatment of multiple organ diseases. This manuscript is well structured overall, and it has been found to add to current research, but it needs little modification that I have outlined in the comments below.
1. Line 27: Please correct the spelling “ceucalyptus” to “Eucalyptus”.
2. Line 39-40: “Research shows that high concentration MA will not cause any discomfort and death when treating animal disease model” please correct grammar.
3. Line 76: being? please fix this word.
4. Line 122: “blue reaction”? or blue color? Please fix this.
5. Line 128: “beat”? please check the word.
6. Please be consistent with the space between temperature range and degree Celsius (I see authors uses both i.e., 60-70 °C and 60-70°C).
7. Line 123: “drop the sample chloroform” please correct grammar.
8. Section 2.3: Line 125, 131 “The sample is MA” please correct grammar.
9. Line 133: “If the sample is heated slightly to light red or purplish red”, authors have to explain what that means.
10. Line 226: Please explain what is “OOs”.
11. Line 573: Recent years has been repeated twice in the same line, the authors have to fix it.
12. Please add these references: DOI: 10.2174/1389450122666210308111159, DOI: 10.1186/s13568-020-01035-1.
13. Similar review on Maslinic acid was published in the year 2014, please cite this journal and the authors needs to explain how there review is important or different. (doi:10.3390/molecules190811538).
14. The authors have to proofread the manuscript carefully and correct any grammar-related mistakes before submission.
Author Response
- Line 27: Please correct the spelling “ceucalyptus” to “Eucalyptus”.
Response: Dear reviewer, thank you for the comment. We have changed it as your guide in 26 line of the revised manuscript.
- Line 39-40: “Research shows that high concentration MA will not cause any discomfort and death when treating animal disease model” please correct grammar.
Response: Dear reviewer, thank you for the kind suggestion. We have revised it as follows: “Research showed that high concentration of MA will not cause any discomfort and death in the treatment of animal models” in 39,40 line of the revised manuscript.
- Line 76: being? please fix this word.
Response: Dear reviewer, thank you for your valuable comment. We have fixed it to "ultrasonic assisted extraction was developed and used" in 85 line of the revised manuscript
- Line 122: “blue reaction”? or blue color? Please fix this.
Response: Dear reviewer, thank you again for this comment. We have fixed it to “blue” in 138 line of the revised manuscript.
- Line 128: “beat”? please check the word.
Response: Dear reviewer, thank you for the comment. We have fixed it to “the chloroform or ethanol solution of the sample is dropped on the filter paper” in 145-146 line of the revised manuscript.
- Please be consistent with the space between temperature range and degree Celsius (I see authors uses both i.e., 60-70 °C and 60-70°C).
Response: Dear reviewer, thank you for this good comment. In the revised manuscript, the space between the temperature range and Celsius has been revised to be consistent
- Line 123: “drop the sample chloroform” please correct grammar.
Response: Dear reviewer, thanks the kind suggestion from the reviewer. We have fixed it to “the sample chloroform solution is dropped on filter paper” in 139 line of the revised manuscript.
- Section 2.3: Line 125, 131 “The sample is MA” please correct grammar.
Response: Dear reviewer, thank you for the comment. We have fixed it to “and this color change could identify the sample as MA” in 141,148 line of the revised manuscript.
- Line 133: “If the sample is heated slightly to light red or purplish red”, authors have to explain what that means.
Response: Dear reviewer, thank you for the kind suggestion. We have revised it to “After the sample solution is heated slightly, if the color changes to light red or purple red, and this color change could identify the sample as MA” in 150-152 line of the revised manuscript.
- Line 226: Please explain what is “OOs”.
Response: Dear reviewer, thank you for the comment. It has been revised it to “olive oil” in 278 line of the revised manuscript.
- Line 573: Recent years has been repeated twice in the same line, the authors have to fix it.
Response: Dear reviewer, thank you for your comment. In the revised manuscript, in line 650 of the revised manuscript, I have deleted a "recent years".
- Please add these references: DOI: 10.2174/1389450122666210308111159, DOI: 10.1186/s13568-020-01035-1.
Response: Dear reviewer, thank you for your comment."DOI: 10.2174/1389450122666210308111159" has been added by me as a revised manuscript reference [16]. “DOI: 10.1186/s13568-020-01035-1” has been added by me as a revised manuscript reference [73].
- Similar review on Maslinic acid was published in the year 2014, please cite this journal and the authors needs to explain how the review is important or different. (doi:10.3390/molecules190811538).
Response:Dear reviewer, thank you for the valuable comment. “doi:10.3390/molecules190811538” has been added by me as a revised manuscript reference [15].
The importance or differences of this review are: “The biological activity of MA has been systematically summarized by scholars, but the extraction of MA and the therapeutic mechanism of MA for various organ diseases have not been discussed[15]. This article will comprehensively and systematically review the research progress in the extraction, separation, identification, bioavailability and bioactivity of MA, and discuss the preventive and therapeutic effects of MA on brain, lung, heart, stomach, intestine, kidney and other organ related diseases.The purpose of writing this review is to provide useful experimental data and research ideas for researchers interested in MA research” in 53-61 line of the revised manuscript.
14.The authors have to proofread the manuscript carefully and correct any grammar-related mistakes before submission.
Response: Dear reviewer, thank you for the comment. We proofread it carefully again.

Reviewer 2 Report
He et al provided a review of Maslinic acid (MA), extraction, purification, identification and, biological activities and briefly discussed its in vivo pharmacokinetics.
After close evaluation of the manuscript. I suggest revision according to the next recommendations and questions:
1- Ceucalyptus in line 27, the (c) should be removed.
2- In lines 42 and 43, in the past few decades, MA has been extracted from a variety of natural plants as people pay more and more attention to new natural compounds that are beneficial to health,
It would be better if the authors can show examples of these plants and their references.
3- In lines 44-46 it would be better to add relevant references as follow, subsequently, researchers studied its pharmacological activity [Ref.], potential molecular mechanism for disease treatment and toxicological effect [Ref.], and found that MA has beneficial effects in medicine [Ref.], health food [Ref.], animal feed [Ref.] and other fields [Ref.].
4- It would be better if the authors include a paragraph in the introduction critically focused on the previous reviews published on the same argument highlighting the difference between themselves and the data discussed in the manuscript.
5- In figure 1 and figure 2, the author should draw Maslinic acid (MA) using Chemdraw for example instead of the used photo.
6- In lines 57-61; Solvent extraction technology is the most classical and convenient extraction method, and also the most commonly used method for extracting natural ingredients. The results showed that hawthorn acid [15-19] could be successfully extracted from olive, apple, jujube and other plants with methanol, ethanol and ethyl acetate. Ignacio Fernandez Pastor [19] et al. determined that the optimal conditions for extracting hawthorn acid from olive were as follows: at T=65-70℃, t=30 min, ethyl acetate (EtOAc) was used as the solvent, and the ratio of material to liquid was 1:40, 1:20 and 1:10 respectively.
These need to be double-cheeked.
7- In lines 67-69 it would be better to add relevant references as follow; In order to improve the extraction efficiency of MA, microwave-assisted extraction (MAE) is widely used in MA extraction [Ref.]. Compared with traditional solvent extraction, MAE has the advantages of low organic solvent consumption, high recovery and low cost [Ref.]. Especially since these sentences are facts.
8- In lines 97-101, the authors compared solvent extraction with microwave-assisted extraction and ultrasonic-assisted extraction, what about the supercritical extraction that is discussed in lines 91-96.
9- In lines 109 and 110; The flow rate of the mobile phase was 1mLmin-1, and the column temperature was set at 30℃. -1 of min needs to be superscript.
10- In section 2.3. I would suggest to the authors to discuss it in two parts, the first is the colorimetric identification and the second is spectroscopic identification. Additionally, in lines 117-133, it would be better to declare a reference for each colorimetric reaction that is discussed in these lines.
11- In Figure 2, I would suggest for the authors to remove the part of the bioactivity from this figure especially since figure 3 discussed Maslinic acid (MA) biological activity.
12- It is very difficult to read the letters and the words in figure 3, this issue needs to be fixed.
13- In line 167, H2O2 need to be corrected as H2O2 (superscript).
14- In lines 170 and 171; MA can lead to the increase of reactive oxygen species in cells at higher doses, further inducing cell damage and apoptosis [38], it would be better if the authors can discuss this critical point with more information.
15- It would be better to rearrange all the therapeutic effects that discussed down the following title as 5. Therapeutic effects of MA.
16- Several references were written in the text in a wrong way, e.g. using a different name from that of the last name of the first author in the references list, also a lot of times et al after the name of the author is missed within the text.
17- In line 404, IC50 need to be corrected as IC50 (superscript).
18- The legend of table 1 is missing.
19- In all the manuscript a space is missed between the end of the text and the Ref. number e.g. (It is reported that MA has antioxidant effect and can improve oxidative damage caused by oxidative stress[26]).
Author Response
- Ceucalyptus in line 27, the (c) should be removed.
Response: Dear reviewer, thank you for the encouraging comments.We have changed it as your guide in 26 line of the revised manuscript.
2-In lines 42 and 43, in the past few decades, MA has been extracted from a variety of natural plants as people pay more and more attention to new natural compounds that are beneficial to health,It would be better if the authors can show examples of these plants and their references.
Response: Dear reviewer, thank you for the kind suggestion. We have fixed it to “MA has been extracted from natural plants, such as jujube, olive and hawthorn [15]” in 49,50 line of the revised manuscript.
3-In lines 44-46 it would be better to add relevant references as follow, subsequently, researchers studied its pharmacological activity [Ref.], potential molecular mechanism for disease treatment and toxicological effect [Ref.], and found that MA has beneficial effects in medicine [Ref.], health food [Ref.], animal feed [Ref.] and other fields [Ref.].
Response: Dear reviewer, thank you again for this comment. I have added references to the revised manuscript, as follows: subsequently, researchers studied its pharmacological activity, potential molecular mechanism for disease treatment and toxicological effect [15], and found that MA has beneficial effects in medicine [15.16], health food [15,16,17], animal feed [18,19] in 52,53 line of the revised manuscript.
4- It would be better if the authors include a paragraph in the introduction critically focused on the previous reviews published on the same argument highlighting the difference between themselves and the data discussed in the manuscript.
Response: Dear reviewer, thank you for the comment. I have added some comments in the revised version about how this review differs from other MA reviews, as follows: “The biological activity of MA has been systematically summarized by scholars, but the extraction of MA and the therapeutic mechanism of MA for various organ diseases have not been discussed[15]. This article will comprehensively and systematically review the research progress in the extraction, separation, identification, bioavailability and bioactivity of MA, and discuss the preventive and therapeutic effects of MA on brain, lung, heart, stomach, intestine, kidney and other organ related diseases. The purpose of writing this review is to provide useful experimental data and research ideas for researchers interested in MA research” in 53-61 line of the revised manuscript.
5- In figure 1 and figure 2, the author should draw Maslinic acid (MA) using Chemdraw for example instead of the used photo.
Response: Dear reviewer, thank you for this good comment. I have used Chemdraw to draw the structural formula of MA.
6- In lines 57-61; Solvent extraction technology is the most classical and convenient extraction method, and also the most commonly used method for extracting natural ingredients. The results showed that hawthorn acid [15-19] could be successfully extracted from olive, apple, jujube and other plants with methanol, ethanol and ethyl acetate. Ignacio Fernandez Pastor [19] et al. determined that the optimal conditions for extracting hawthorn acid from olive were as follows: at T=65-70℃, t=30 min, ethyl acetate (EtOAc) was used as the solvent, and the ratio of material to liquid was 1:40, 1:20 and 1:10 respectively.
These need to be double-cheeked.
Response: Dear reviewer, thank you again for this comment. These have been double-cheeked.
7- In lines 67-69 it would be better to add relevant references as follow; In order to improve the extraction efficiency of MA, microwave-assisted extraction (MAE) is widely used in MA extraction [Ref.]. Compared with traditional solvent extraction, MAE has the advantages of low organic solvent consumption, high recovery and low cost [Ref.]. Especially since these sentences are facts.
Response: Dear reviewer, thank you again for this comment. I have added references to the revised manuscript, as follows: In order to improve the extraction efficiency of MA, microwave-assisted extraction (MAE) is widely used in MA extraction [20,25]. Compared with traditional solvent extraction, MAE has the advantages of low organic solvent consumption, high recovery and low cost [25].
8- In lines 97-101, the authors compared solvent extraction with microwave-assisted extraction and ultrasonic-assisted extraction, what about the supercritical extraction that is discussed in lines 91-96.
Response: Dear reviewer, thank you again for this comment. “Supercritical fluid extraction is the only method that does not consume organic solvent, but it takes a long time and has low efficiency” in 108-110 line of the revised manuscript.
9- In lines 109 and 110; The flow rate of the mobile phase was 1mLmin-1, and the column temperature was set at 30℃. -1 of min needs to be superscript.
Response: Dear reviewer, thank you for the kind suggestion. We have changed it as your guide in 123 line of the revised manuscript.
10- In section 2.3. I would suggest to the authors to discuss it in two parts, the first is the colorimetric identification and the second is spectroscopic identification. Additionally, in lines 117-133, it would be better to declare a reference for each colorimetric reaction that is discussed in these lines.
Response: Dear reviewer, thank you for your valuable comment. The reference of colorimetric reaction is added to the reference of the revised manuscript [35, 36], and its reference comes from our domestic textbooks.
The identification of MA is discussed in two parts, as follows:
“Colorimetric identification
MA is a pentacyclic triterpene acid compound with 2,3 dihydroxy groups, which has the characteristic color reaction of terpenoids. MA is a 12 alkene structure with slow color development. The main color reaction is as follows[35,36]: 1) MA reacts with acetic anhydride concentrated sulfuric acid: dissolve the sample in acetic anhydride and add a few drops of concentrated sulfuric anhydride (1:20), which can produce yellow → red → purple → blue, and finally fade, then the sample can be identified as MA; 2) MA reacts with trichloroacetic acid: the sample chloroform solution is dropped on filter paper, then 1 drop of 25% TCA solution is added and heated to 100°C for color development, if red to purple color was generated, and this color change could identify the sample as MA; 3) MA reacts with chloroform concentrated sulfuric acid: the sample is dissolved in chloroform. After adding concentrated sulfuric acid, chloroform is red or blue, and the sulfuric acid layer is green fluorescent, so the sample can be identified as MA; 4) MA reacts with antimony pentachloride: the chloroform or ethanol solution of the sample is dropped on the filter paper, and then 20% antimony pentachloride chloroform solution is sprayed on the filter paper. If it is heated, it will show blue, gray blue and gray purple color spots, and this color change could identify the sample as MA; 5) Glacial acetic acid-acetyl chloride reaction: The sample is dissolved in glacial acetic acid, and then a few drops of acetyl chloride and a few grains of zinc oxide are added. After the sample solution is slightly heated, if the color changed to light red or purplish red, and this color change could identify the sample as MA.” in 131-152 line of the revised manuscript.
“Chromatography
In addition to colorimetric methods, we can also use chromatographic methods to identify MA. Guo, S. et al. used high performance liquid chromatography (HPLC) coupled with evaporative light scattering detection (ELSD) to identify MA in date palm leaves [6]. Yang Jie et al. also analyzed the content of hawthorneic acid in plum blossoms using high performance liquid chromatography coupled with evaporative light scattering detection (HPLC-ELSD) [37]. Perez-Camino M C et al. used gas chromatography (GC) to alkalize hawthorne acid samples and then injected them into a gas chromatograph to obtain gas chromatograms of alkylated derivatives of hawthorne acid for the identification of MA in olive oil [38]. In GC, due to its low volatility and high molecular weight, MA cannot be injected directly into the chromatograph and must be derivatized prior to gas chromatographic analysis. The use of HPLC hinders the detection of this compound because it has a saturated backbone, does not show fluorescence, and has very low UV absorption. Therefore, these methods may not be sensitive enough to detect low doses of hawthorneic acid. Sanchez-Gonzalez Marta developed a new detection technique, liquid chromatography-atmospheric pressure chemical ionization mass spectrometry (LC-APCI-MS), for the analysis of hawthorneic acid in plasma [39]. Compared to GC, LC-APCI-MS does not depend on specific groups in the molecule, thus reducing the derivatization step and avoiding sample consumption. Also, it can be used for highly specific detection without long chromatographic separations compared to HPLC. Therefore, LC-APCI-MS may be a relatively efficient and sensitive method for the determination of MA concentration.” in 154-176 line of the revised manuscript.
11- In Figure 2, I would suggest for the authors to remove the part of the bioactivity from this figure especially since figure 3 discussed Maslinic acid (MA) biological activity.
Response: Dear reviewer, thank you again for this comment. In Figure 2, the content of biological activity is deleted, and the relevant methods of MA chromatographic identification are added.
12- It is very difficult to read the letters and the words in figure 3, this issue needs to be fixed.
Response: Dear reviewer, thank you for the kind suggestion. Figure 3 has been fixed by me.
13- In line 167, H2O2 need to be corrected as H2O2 (superscript).
Response: Dear reviewer, thank you for the encouraging comments. We have changed it as your guide in 208 line of the revised manuscript.
- In lines 170 and 171; MA can lead to the increase of reactive oxygen species in cells at higher doses, further inducing cell damage and apoptosis [38], it would be better if the authors can discuss this critical point with more information.
Response: Dear reviewers, thank you for your valuable comments. My discussion on this point is as follows: “The results showed that in MA treated B16F10 melanoma cells, ROS levels were decreased at MA concentrations less than (IC50/8) and increased at MA concentrations greater than (IC50/8) [50]” in 213-215 line of the revised manuscript.
15- It would be better to rearrange all the therapeutic effects that discussed down the following title as 5. Therapeutic effects of MA.
Response: Dear reviewer, thank you for the encouraging comments.We have changed it as your guide.
16- Several references were written in the text in a wrong way, e.g. using a different name from that of the last name of the first author in the references list, also a lot of times et al after the name of the author is missed within the text.
Response: Dear reviewer, thank you for the encouraging comments. We have changed it as your guide. All the names mentioned in the article have been fixed by referring to the name of the first author in the reference list.
17- In line 404, IC50 need to be corrected as IC50 (superscript).
Response: Dear reviewer, thank you for the encouraging comments.We have changed it as your guide in 466 line of the revised manuscript.
18- The legend of table 1 is missing.
Response: Dear reviewers, thank you for your valuable comments. The legend in Table 1 has been added.
19- In all the manuscript a space is missed between the end of the text and the Ref. number e.g. (It is reported that MA has antioxidant effect and can improve oxidative damage caused by oxidative stress[26]).
Response: Dear reviewer, thank you for the kind suggestion. The space between the end of the text and the references has been added by me.

Round 2
Reviewer 2 Report
The authors have already made reasonable efforts in the revised version, just some minors points need to be done before being published:
1- In maslinic acid structure, the geminal methyl angle must be corrected as shown in the attached file.
2- In line 65 hawthorn acid need to be checked and if wrong to be corrected as maslinic acid.
3- In line 90, Xie, P. [20]et al. need to be corrected Xie et al. [20], please check to be all the references used in that way.
4- [35,36] in line 130, it would be better to be at the end of the text in line 147.
5- In line 170 In conclusion, the extraction, purification, identification and biological activity of MA are shown in Figure 2
Biological activity should be removed since it does not exist anymore in figure 2 and should be removed from the legend of Figure 2 in line 174 as well.
6- I think that the main title 3........ is missing before subtitle 3.1. Hypoglycemic effect in line 175.
7- Several references are missing et al after the author name, for example, Hsia, T. C. [61] in line 375, please double-check in the whole manuscript and use the same way as mentioned in point No. 3.

Author Response
1- In maslinic acid structure, the geminal methyl angle must be corrected as shown in the
attached file.
Response: Dear reviewer, thank you for the comment. We have changed it as your
guide in 42 line of the revised manuscript.
2- In line 65 hawthorn acid need to be checked and if wrong to be corrected as maslinic acid.
Response: Dear reviewer, thank you for the kind suggestion.We have fixed
“hawthorn acid” to “MA” in 62 line of the revised manuscript. (in purple)
3- In line 90, Xie, P. [20]et al. need to be corrected Xie et al. [20], please check to be all the
references used in that way.
Response: Dear reviewer, thank you again for this comment. All references have
been changed to your guide in the revised manuscript. (in purple)
4- [35,36] in line 130, it would be better to be at the end of the text in line 147.
Response: Dear reviewer, thank you for the good comment. We have changed it as
your guide. We have placed [35,36] in line 150 of the revised manuscript. (in
purple)
5- In line 170 In conclusion, the extraction, purification, identification and biological activity
of MA are shown in Figure 2 Biological activity should be removed since it does not exist
anymore in figure 2 and should be removed from the legend of Figure 2 in line 174 as well.
Response: Dear reviewer, thank you for the valuable comment. We have changed
it as your guide.We have removed the biological activity in line 173,175 of the
revised manuscript. (in purple)
6- I think that the main title 3........ is missing before subtitle 3.1. Hypoglycemic effect in line
Response: Dear reviewer, thank you again for the good comment. We have
changed it as your guide. “3. Biological activity of MA” has been added in line 176
of the revised manuscript. (in purple)
7- Several references are missing et al after the author name, for example, Hsia, T. C. [61] in
line 375, please double-check in the whole manuscript and use the same way as mentioned
in point No. 3.
Response: Dear reviewer, thank you for the encouraging comments. All references
have been changed to your guide in the revised manuscript. (in purple)
